# The Effect of Some Wild Grown Plant Extracts and Essential Oils on *Pectobacterium betavasculorum*: The Causative Agent of Bacterial Soft Rot and Vascular Wilt of Sugar Beet

**DOI:** 10.3390/plants11091155

**Published:** 2022-04-25

**Authors:** Mina Rastgou, Younes Rezaee Danesh, Sezai Ercisli, R. Z. Sayyed, Hesham Ali El Enshasy, Daniel Joe Dailin, Saleh Alfarraj, Mohammad Javed Ansari

**Affiliations:** 1Department of Plant Protection, Faculty of Agriculture, Urmia University, Urmia 5756151818, Iran; y.rdanesh@yahoo.com; 2Department of Horticulture, Agricultural Faculty, Ataturk University, Erzurum 25240, Turkey; sercisli@atauni.edu.tr; 3Department of Entomology, Asian PGPR Society for Sustainable Agriculture, Auburn University, Auburn, AL 36830, USA; sayyedrz@gmail.com; 4Department of Microbiology, PSGVP Mandal’s Shri S I Patil Arts, G B Patel Science and STKVS Commerce College, Shahada 425409, India; 5Institute of Bioproduct Development (IBD), Universiti Teknologi Malaysia (UTM), Johor 81310, Malaysia; henshasy@ibd.utm.my (H.A.E.E.); jddaniel@utm.my (D.J.D.); 6School of Chemical and Energy Engineering, Faculty of Engineering, Universiti Teknologi Malaysia (UTM), Johor 81310, Malaysia; 7City of Scientific Research and Technology Applications (SRTA), New Borg Al Arab 21934, Egypt; 8Zoology Department, College of Science, King Saud University, Riyadh 11451, Saudi Arabia; salfarraj@hotmail.com; 9Department of Botany, Hindu College, Mahatma Jyotiba Phule Rohilkhand University Bareilly, Moradabad 244001, India; mjavedansari@gmail.com

**Keywords:** antibacterial, essential oil, plant extract, sugar beet, *Pectobacterium betavasculorum*

## Abstract

The bacterial soft rot and vascular wilt of sugar beet are the major diseases of sugar crops globally induced by *Pectobacterium* *betavasculorum* and *P. carotovorum* subsp. *carotovorum* (*Pcc*). The control of this bacterial disease is a severe problem, and only a few copper-based chemical bactericides are available for this disease. Because of the limitations of chemicals to control plant bacterial pathogens, the essential oils and extracts have been considered one of the best alternative strategies for their control. In this study, twenty-seven essential oils and twenty-nine plant extracts were extracted and evaluated for their antibacterial activities against *Pectobacterium betavasculorum* isolate C3, using the agar diffusion method at 0.01%, 0.1%, and 100% (*v*/*v*). Pure *Pimpinella anisum* L. oil exhibited the most anti-bacterial activity among three different concentrations of essential oils and extracts, followed by *Thymus vulgaris* L. oil and *Rosa multiflora* Thunb. extract. The efficacy of effective essential oils and extracts on Ic_1_ cultivar of sugar beet seeds germination and seedling growth in vivo also were tested. The seed germination of the Ic1 cultivar was inhibited at all the concentrations of essential oils used. Only extracts of *Rosa multiflora* Thunb., *Brassica oleracea* L., *Lactuca serriola* L., *Salvia rosmarinus* Spenn., *Syzygium aromaticum* (L.) Merr. and L.M.Perry, *Eucalyptus globulus* Labill., and essential oils of *Ocmium basilicum* L., *Pimpinella anisum* L., and *Mentha× piperita* L.L. in 0.1% concentration had no inhibition on seed germination and could improve seedling growth. This is the first report of the antibacterial activity of essential oils and extracts on *Pectobacterium betavasculorum*.

## 1. Introduction

Bacterial sugar beet soft rot disease caused mainly by *Pectobacterium betavasculorum* and *P. carotovorum* subsp. *carotovorum* (*Pcc*) (formerly *Erwinia carotovora* ssp. *carotovora*). The genus *Pectobacterium* is an important Gram-negative plant pathogen that belongs to the family Enterobacteriaceae. This genus causes severe losses of many commercial crops, including sugar cane, in the field and during storage [1,2,3]. It is proven that *Pectobacterium betavasculorum* (*Pb*) is the causal agent of sugar beet soft rot and vascular wilt disease in Iran [4]. The control of bacterial diseases is a severe problem in plant production. Only a few chemical bactericides are available for management using copper-based bactericidal substances or different antibiotics [5,6,7]. It is shown that chemical bactericides have limitations such as general toxicity, the development of resistant bacterial strains, and negative effects on the environment and the yield because of accumulation in the food chain and long degradation periods. Different plant compounds have been tested for antimicrobial activity as an alternative strategy for plant protection against pathogens, including bacteria. Essential oils, as odorous and volatile products of the secondary metabolism of aromatic plants, normally formed in special cells or groups of cells, could be used as antimicrobial agents in the control of the disease caused by bacteria [8,9] and were proved to be better substitutes for synthetic antibiotics as they have an easily biodegradable nature [10]. Their antimicrobial activity is assigned to small terpenoids and phenolic compounds [11]. The inhibitory effect of some essential oils has been demonstrated on some plant pathogenic bacteria, including *Clavibacter michiganensis* subsp. *michiganensis* [12,13], *Pectobacterium carotovorum* [14,15,16,17,18,19,20], *Pseudomonas mariginalis* pv. *mariginalis*, *Erwinia chrysantemi* [21] *Ralstonia solanacearum* [14], *Xantomonas campestris* pv. *malvacearum*, *Pseudomonas syringae* pv. *tomato* [13], *Pseudomonas syringae* pv. *syringae*, *Xanthomonas arboricola* pv. *pruni* [22], *Xanthomonas oryzae*, *Xanthomonas malvacearum* [23], *Erwinia amylovora* [19,24,25,26], *Dickeya solani*, *Agrobacterium tumefaciens* [24,25,26], *Pseudomonas aeruginosa* [27], *Xanthomonas oryzae* pv. *oryzae*, *X. oryzae* pv. *oryzicola* [28], *Pseudomnas putida* and *Pseudomonas fluorescens* [29]. Plant extracts from different organs of plants, including roots, barks, seeds, shoots, leaves, and fruits, were studied for anti-bacterial abilities against plant pathogenic bacteria [30,31,32,33,34,35,36]. The interest in studying the antimicrobial activities of essential oils and plant extracts for plant disease control, especially bacterial diseases, has recently increased [23]. The present work aimed to evaluate the antibacterial activities of twenty-seven essential oils and twenty-nine plant extracts on *Pectobacterium betavasculorum*; the causal agent of bacterial soft rot and vascular wilt of sugar beet in Iran. We also tested the efficacy of these essential oils and extracts on seed germination and stem and root growth of sugar beet seedlings. The detailed information on the anti-bacterial effect of these essential oils and extracts on, *Pectobacterium betavasculurum* is scarce.

## 2. Materials and Methods

### 2.1. Extraction

#### 2.1.1. Essential Oils

The essential oil of all 27 air-dried plants (100 g) was extracted by hydro-distillation for 4–6 h using a Clevenger apparatus [37]. The distillate oils were dried over anhydrous sodium sulfate (Na_2_SO_4_) and stored in closed dark vials at 4 °C. This extract was serially diluted with sterilized distilled water. The plants were used for extraction are summarized in Table 1.

#### 2.1.2. Plant Extracts

One-hundred grams of all the 29 plant leaves were powdered and extracted separately in ethanol by soxhlet extraction [38]. Extraction was done for 18–24 h. Ethanol was evaporated in a water bath. A 5 g of each plant was soaked in 95 mL distilled water for 1 h at room temperature for aqueous extracts. Each extract was serially diluted with sterilized water. The plants were used for extraction are summarized in Table 2. Leaves were used to extract almost all plants, and flowers and leaves were used for *Robinia pseudocacacia* L. and *Satureja hortensis* L. All the plant materials used in the present study were kindly supplied by the Medicinal and Aromatic Plants collection of the Department of Horticulture, College of Agriculture, Urmia University, Urmia, Iran.

### 2.2. Bacterial Strain

*Pectobacterium betavasculorum* strain C3, the causal agent of sugar beet soft rot and vascular wilt disease in Iran, was obtained from an Iranian collection (Department of Plant Protection, College of Agriculture, Kerman University, Kerman, Iran). Bacterial culture was maintained on Luria–Bertani (LB) agar medium at 4 °C.

### 2.3. In Vitro Antibacterial Activity

The well diffusion method determined the in vitro antibacterial activity of the essential oils and plant extracts. Petri plates with 15 mL of nutrient agar (0.5% peptone, 0.3% beef extract/yeast extract, 1.5% bacteriological agar, 0.5% NaCl, Merck) were inoculated with 1.5 × 10^7^ CFU/mL of 24 h suspensions of *Pectobacterium betavasculorum*, grown on nutrient agar. Wells were cut into the agar and filled with 20 μL of each essential oil or extract. The plates were placed for 2 h at 4 °C to allow the diffusion of essential oils and extracts and cultivatedfor 24–48 h at 26 °C. After cultivation, the distinct growth inhibition zone around the wells was measured in mm. The antibacterial activity was assessed by measuring the diameter of the inhibition zone in mm. The experiments were performed in four replicates and 0.01% (10^−2^), 0.1% (10^−1^), and 100% solutions of the essential oils and extracts in absolute ethanol [29,30,31]. 

### 2.4. In Vivo Test

#### 2.4.1. On Seed Germination

Sugar beet seeds of Ic_1_ cultivar susceptible to *Pectobacterium betavasculorum* were disinfested by emerging in 1% NaOH for 5 min, then rinsed and washed in sterilized water and air-dried at 24 °C according to [31]. Disinfected seeds were soaked for 30 s individually in two concentrations of oils and extracts (100% and 0.1) and then laid between two wet sterile Whatman filter papers in a Petri dish for 10 days at 26 °C, and 500 μL water was added to each petri dish daily. Petri dishes were checked daily to remove the germinated seeds and determine the negative effect of the oils and extracts on seed germination. This experiment was done in three replicates with 30 seeds for each replicate. In preliminary experiments, it was observed that several used essential oils and extracts caused phytotoxicity and inhibited seeds germination. To remove these negative adverse effects and statistical analysis with low errors, the negative efficacy of the essential oils and extractswas measured.

#### 2.4.2. On Stem and Root Growth Rate of Seedling

After surface sterilization, sugar beet seeds of the Ic1 cultivar were soaked in essential oils and extracts for 30 s. For soil infestation, *Pectobacterium betavasculorum* (1 × 10^1^ cfu/g) was added and mixed thoroughly to ensure an equal distribution of *Pectobacterium betavasculorum* inoculum. Infested soil was then filled in sterile glass containers and irrigated daily for one week before sowing. Seeds were sown in infested soil and kept at 28–30 °C and relative humidity between 70–90% until the end of the experiment. Approximately five weeks after sowing, the seedling stem and root growth rate were measured. This experiment was done in four replicates. All tests were laid in Completely Randomized Design (CRD) with four replicates. Glass containers filled with sterilized soil and sterilized seeds were used as a negative control, and glass containers filled with *Pectobacterium betavasculorum-*infested soil and sterilized seeds were used as a positive control.

#### 2.4.3. Statistical Analysis

All the experiments were performed in triplicates, Data was statistically analyzed, and mean values were calculated. Data were subjected to Analysis of Variance (ANOVA) using an SAS 2012 model (SAS Version 9.2., SAS Institute, Cary, NC, USA). Effects were declared significant or non-significant at (*p* ≤ 0.01) level of probability for those parameters where the ANOVA was significant using least significant difference (LSD) at 1% level of probability (*p* ≤ 0.01) according to Duncan’s multiple range test.

## 3. Results

### 3.1. In Vitro Test

The anti-bacterial activities of 27 essential oils, 29 plant extracts on *Pectobacterium betavasculorum* showed significant differences among essential oils and extracts in inhibiting bacterial growth (*p* ≤ 0.01). Essential oils of *Pimpinella anisum* L., *Thymus vulgaris* L., *Satureja hortensis* L., *Syzygium aromaticum*, *Eucalyptus globulus* Labill., *Ocimum basilicum* L., *Mentha pulegium* L., *Rosmarinus officinalis*, *Junipers polycarpus*, *Mentha× piperita* L., and *Artemisia dracunculus* L. not Hook.f. 1881 with 12.87, 9.25, 6.62, 6.37, 5.87, 5.25, 4.87, 4.12, 4.00, 3.25, and 1.12 mm inhibition zone (*p* ≤ 0.01) respectively, almost inhibit *Pectobacterium betavasculorum* at the pure concentration. Still, only essential oils of *Pimpinella anisum* L., *Thymus vulgaris* L., and *Syzygium aromaticum* with 5.08, 4.92, and 4.67 mm inhibition zone (*p* ≤ 0.01) had inhibition at 10^−1^ concentration, and no essential oil inhibited bacterial growth at 10^−2^ concentration compared with the control (Table 3). The other 15 essential oils showed no anti-bacterial activities on *Pectobacterium betavasculorum* at any concentration. Extracts of *Rosa multiflora* Thunb., *Lactuca serriola* L., *Brassica oleracea* L., *Eucalyptus globulus* Labill., *Syzigium aromaticum*, and *Salvia rosmarinus* Spenn. with 7.25, 4.5, 3.87, 3.62, 3.5, and 1.37 mm inhibition zone in pure concentration, respectively, and 3.3, 2, 1.7, 1.6, 1.3, and 0 mm at 10^−1^ concentration and no inhibition on bacterial growth at 10^−2^ concentration compared with the control was observed (Table 3). The other 22 extracts listed in Table 1 showed no anti-bacterial activities on *Pectobacterium betavasculorum* at any concentration.

### 3.2. In Vivo Test

All used essential oils and extracts had phytotoxicity on seeds at pure concentration, and no seed was germinated at this concentration. The statistical analysis showed a significant difference between treatments with control (*p* ≤ 0.01). At 10^−1^ concentration, essential oils of *Ocimum basilicum* L., *Pimpinella anisum* L., and *Mentha× piperita* L. had 21.34, 19.78, and 15.52% negative efficacy on seed germination, respectively. Other essential oils inhibited seed germination completely (Table 4). Among plant extracts, *Rosa multiflora* Thunb., *Brassica oleracea* L., *Lactuca serriola* L., *Salvia rosmarinus* Spenn., *Syzigium aromaticum*, and *Eucalyptus globulus* had 74.1, 72.2, 70.3, 66.5, 56.05, and 56.05% negative efficacy on seed germination, respectively. So only these essential oils and extracts were used to measure the growth rate of seedling stem and root in the next step. They had a significant difference in seedling growth than the positive control (*p* ≤ 0.01). There was no statistically significant difference between these essential oils and extracts to improve the seedling’s root and stem growth (Table 5, Figure 1).

## 4. Discussion

The various essential oils and extracts exhibited varying degrees of inhibition against the C3 isolate of *Pectobacterium betavasculorum* pathogen, the causal agent of soft rot and vascular wilt of sugar beet in Iran [4], using the well diffusion method in vitro. In our study, anise (*Pimpinella anisum* L.) essential oil had the most potent inhibitory effect in vitro test (*p* ≤ 0.01). Thyme (*Tymus vulgaris* L.) could inhibit *Pectobacterium betavasculorum* growth in vitro, but in vivo test, it was phytotoxic and had negative efficacy on seed germination. Some researchers reported the antibacterial activity of thyme on *Clavibacter michiganensis* subsp. *michiganensis* [12], *Pectobacterium carotovorum* [14,15,16], *Pseudomonas mariginalis* pv. *mariginalis*, *Erwinia chrysantemi* [22], *Xanthomonas vesicatoria* [39], Pseudomonas syringae pv. tomato [40], and *Ralstonia solanacearum* [14]. It has stronger antimicrobial properties as its oil possesses more phenolics, including carvacrol, eugenol, and thymol. They act through the Quorum sensing machinery to inhibit specific virulence determinants, including biofilm formation and plant cell wall degrading enzymes (PCWDEs) [41]. The results of the present study are in agreement with the findings of earlier researchers that some essential oils, including thyme oil, are highly phytotoxic, especially on seed germination [42,43,44]. Among other tested essential oils, peppermint (*Mentha× piperita* L.) and sweet basil (*Ocimum basilicum* L.) had good inhibitory effects in the in vitro test. Also, they had a positive effect on seed germination and seedling growth under in vivo tests. *Rosa multiflora* Thunb. extract had a good inhibitory effect on *Pectobacterium betavasculorum* in vitro test and seed germination in vivo. To our knowledge, this is the first study to provide data on the evaluation of essential oils and plant extracts against *Pectobacterium betavasculorum*. According to the in vitro and in vivo tests, anise (*Pimpinella anisum* L.), peppermint (*Mentha× piperita* L.) and sweet basil (*Ocimum basilicum* L.) essential oils and Rosa (*Rosa multiflora* Thunb.), cabbage (*Brassica oleracea* L.), prickly lettuce (*Lactuca serriola* L.), Rosemary (*Salvia rosmarinus* Spenn.), clove (*Syzigium aromaticum*) and Eucalyptus (*Eucalyptus globulus* Labill.) extracts have a strong anti-bacterial activity on *Pectobacterium betavasculorum*. Rosemary essential oil has been tested on *Clavibacter michiganensis* subsp. *Michiganensis* [12], *Ralstonia solanacearum* [14], *Pseudomonas mariginalis* pv. *mariginalis*, *Erwinia chrysantemi* [20], and *Pectobacterium carotovorum* [14,16], but no report on its extract anti-bacterial activity exists. This study revealed that essential oils and extracts of these plants have anti-bacterial activities and can be used to control *Pectobacterium betavasculorum*. The treatment with these essential oils and extracts of sugar beet seeds artificially contaminated with *Pectobacterium betavasculorum* confirmed the efficacy determined in the in vitro experiments. The anti-bacterial activity of these plants may be attributed to various antimicrobial phytochemicals [45,46] and so should be studied clearly. This study has shown that some plants effectively inhibit *Pectobacterium betavasculorum* growth. Most of the plants here were tested for their antibacterial activity first time. The antimicrobial activity evaluated in this work could be attributed to different compounds in variable amounts in plant essential oils and extracts. These compounds should be studied in detail. It is proven that the assayed antimicrobial activity of the plant species depends on the botanical species, age, part of the plant studied, and the solvent used for the extraction procedures [46]. Further analyses on the concentrations and combinations of these essential oils and extracts are required to obtain an anti-bacterial effect on *Pectobacterium betavasculorum* with minimum undesirable effects on seed germination and seedling growth. They should also be used in field studies to learn their effectiveness in field conditions. Therefore, more research on the activity of these plants against the other plant pathogenic bacteria and other plant pathogens would be of great value. These findings persuaded us to continue screening more plant species for their anti-bacterial activity and their compounds in agriculture.In conclusion, this research evaluates the effectiveness of anise (*Pimpinella anisum* L.), peppermint (*Mentha× piperita* L.) and sweet basil (*Ocimum basilicum* L.) essential oils, and Rosa (*Rosa multiflora* Thunb.), cabbage (*Brassica oleracea* L.), prickly lettuce (*Lactuca serriola* L.), Rosemary (*Salvia rosmarinus* Spenn.), clove (*Syzigium aromaticum*) and Eucalyptus (*Eucalyptus globulus* Labill.) extracts on *Pectobacterium betavasculorum*; the cusal agent of bacterial soft rot and vascular wilt of sugar beet in Iran. This research would augment the growing body of plant pathogenic *Pectobacterium* research and its utility in management studies of this pathogen using natural and non-chemical materials.

## Figures and Tables

**Figure 1 plants-11-01155-f001:**
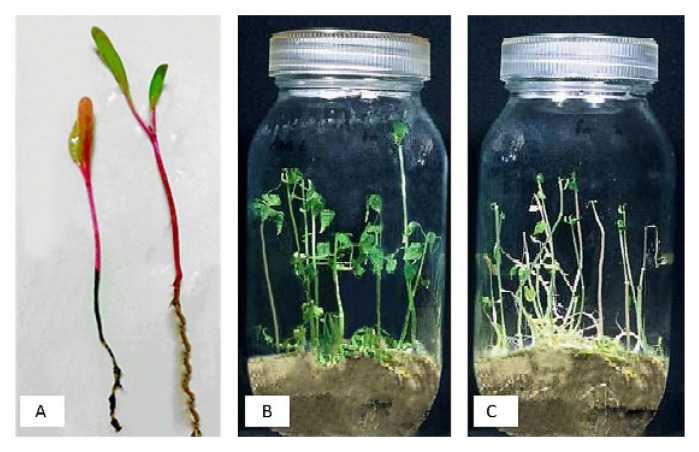
In-vivo test on seedling growth, (**A**) root and stem growth, (**B**) negative control, (**C**) positive control. Glass containers filled with sterilized soil and sterilized seeds were used as a negative control, and glass containers filled with *Pectobacterium betavasculorum-*infested soil and sterilized seeds used as a positive control.

**Table 1 plants-11-01155-t001:** Essential oils extracted by hydro-distillation and tested for antibacterial activity against *P. betavasculorum*.

Essential Oils		
Family Name	Scientific Name	Common Name
Lamiaceae	*Thymus vulgaris* L.	Thyme
Lamiaceae	*Ocimum basilicum* L.	Sweet basil
Lamiaceae	*Salvia rosmarinus* Spenn.	Common Rosemary
Lamiaceae	*Mentha× piperita* L.	Peppermint
Apiaceae	*Foeniculum vulgare* Mill.	Fennel
Apiaceae	*Cuminum cyminum* L.	Cumin
Asteraceae	*Filifolium sibiricum* (L.) Kitam.	Artemisia
Lamiaceae	*Origanum majorana* L.	Marjoram
Apiaceae	*Heracleum persicum* Desf. ex Fisch.	Golpar
Myrtaceae	*Syzygium aromaticum* (L.) Merr. & L.M.Perry	Clove
Apiaceae	*Anethum graveolens* L.	Dill
Rutaceae	*Citrus × sinensis* (L.) Osbeck	Orange
Asteraceae	*Achillea millefolium* L.	Yarrow
Lamiaceae	*Satureja hortensis* L.	Summer savory
Apiaceae	*Pimpinella anisum* L.	Anise
Myrtaceae	*Eucalyptus globulus* Labill.	Eucalyptus
Lamiaceae	*Mentha pulegium* L.	Pennyroyal
Cupressaceae	*Juniperus polycarpos* K.Koch	Juniper
Asteraceae	*Artemisia dracunculus* L. not Hook.f. 1881	Tarragon
Lamiaceae	*Teucrium polium* L.	Felty germander
Piperaceae	*Piper nigrum* L.	Black pepper
Asteraceae	*Chrysanthemum indicum* L.	Chrysanthemum
Apiaceae	*Petroselinum crispum* (Mill.) Fuss	Parsley
Myristicaceae	*Myristica fragrans* Houtt.	Nutmeg
Rutaceae	*Citrus reticulata* Blanco, 1837	Mandarin orange
Lamiaceae	*Lavandula officinalis* Mill.	Lavender
Solanaceae	*Nicotiana tabacum* L.	Tobacco

**Table 2 plants-11-01155-t002:** Plant extracts extracted by soxhlet extraction method and tested for antibacterial activity against *P. betavasculorum*.

Plant Extracts		
Family Name	Scientific Name	Common Name
Lamiaceae	*Thymus vulgaris* L.	Thyme
Lamiaceae	*Ocimum basilicum* L.	Sweet basil
Lamiaceae	*Salvia rosmarinus* Spenn.	Common Rosemary
Lamiaceae	*Mentha× piperita* L.	Peppermint
Asteraceae	*Lactuca serriola* L.	Prickly lettuce
Rosaceae	*Rosa multiflora* Thunb.	Hybrid tea rose
Ranunculaceae	*Staphisagria macrosperma* Spach, 1839	Larkspur
Apiaceae	*Ca sativum* L.	Coriander
Rosaceae	*Chaenomeles japonica* (Thunb.) Lindl. ex Spach	Japanese quince
Fabaceae	*Caesalpinia pulcherrima* (L.) Sw.	Poinciana
Myrtaceae	*Syzygium aromaticum* (L.) Merr. & L.M.Perry	Clove
Apocynaceae	*Nerium oleander* L.	oleander
Fabaceae	*Robinia pseudoacacia* L.	Black Locust
Lamiaceae	*Satureja hortensis* L.	Summer savory
Apiaceae	*Pimpinella anisum* L.	Anise
Myrtaceae	*Eucalyptus globulus* Labill.	Eucalyptus
Lamiaceae	*Mentha pulegium* L.	Pennyroyal
Cupressaceae	*Juniperus polycarpos* K.Koch	Juniper
Asteraceae	*Artemisia dracunculus* L. not Hook.f. 1881	Tarragon
Elaeagnaceae	*Elaeagnus angustifolia* L.	Russian olive
Euphorbiaceae	*Ricinus communis* L.	Castor bean
Amaranthaceae	*Amaranthus retroflexus* L.	Common Amaranth
Apiaceae	*Petroselinum crispum* (Mill.) Fuss	Parsley
Lamiaceae	*Mentha aquatica* L.	Water mint
Brassicaceae	*Brassica oleracea* L.	Ornamental cabbage
Solanaceae	*Nicotiana tabacum* L.	Tobacco
Apiaceae	*Eryngium maritimum* L	Eryngo
Amaranthaceae	*Halostachys belangeriana* (Moq.) Botsch.	Halostachys

**Table 3 plants-11-01155-t003:** Antibacterial activity of essential oils and extracts in vitro on *Pectobacterium betavasculorum*.

Diameter of the Inhibition Zone (mm) in 10-2 Concentration	Diameter of the Inhibition Zone (mm) in 10-1 Concentration	Diameter of the Inhibition Zone (mm) in Pure Concentration	Plant
0	0	0	Negative control (sterilized water)
0	5.08 ± 0.49 ^a^	12.87 ± 0.83 ^a^	*Pimpinella anisum* L. Eo
0	4.92 ± 0.73 ^a^	9.25 ± 1.25 ^a^	*Thymus vulgaris* L. Eo
0	3.3 ± 0.27 ^ab^	7.25 ± 1.03 ^bc^	*Rosa multiflora* Thunb. Ex
0	0 ^c^	6.62 ± 0.52 ^bc^	*Satureja hortensis* L. Eo
0	4.67 ± 0.4 ^a^	6.37 ± 0.74 ^bc^	*Syzygium aromaticum* Eo
0	0 ^c^	5.87 ± 0.58 ^bc^	*Eucalyptus globulus* Labill. Eo
0	0 ^c^	5.25 ± 0.46 ^c^	*Ocimum basilicum* L. Eo
0	0 ^c^	4.87 ± 0.64 ^c^	*Mentha pulegium* L. Eo
0	2 ± 0.5 ^b^	4.5 ± 1.06 ^c^	*Lactuca serriola* L. Ex
0	0 ^c^	4.12 ± 1.12 ^c^	*Salvia rosmarinus* Spenn. Eo
0	0 ^c^	4.00 ± 0.75 ^c^	*Junipers polycarpus* Eo
0	1.7 ± 0.27 ^b^	3.87 ± 0.83 ^d^	*Brassica oleracea* L. Ex
0	1.6 ± 0.22 ^b^	3.62 ± 0.74 ^d^	*Eucalyptus globulus* Labill. Ex
0	1.3 ± 0.27 ^b^	3.5± 0.53 ^d^	*Syzygium aromaticum* Ex
0	0 ^c^	3.25 ± 0.03 ^d^	*Mentha× piperita* L. Eo
0	0 ^c^	1.37 ± 0.35 ^e^	*Rosmarinus officinalis* Ex
0	0 ^c^	1.12 ± 1.35 ^e^	*Artemisia dracunculus* L. not Hook.f. 1881 Eo

Eo = Essential oil, Ex = Plant extract. Values are the mean diameter of inhibition zone (mm) ± SD of four replications, followed by different letters in columns are significantly different (*p* ≤ 0.01).

**Table 4 plants-11-01155-t004:** Effect of essential oils and extracts on seed germination.

Negative Efficacy ^2^ (%)	No. of Germinated Seeds ^2^ (30)	Negative Efficacy ^1^ (%)	No. of Germinated Seeds ^1^ (30)	Treatment
83.3 ^a^	25.75 ± 1.71	83.3 ^a^	25.75 ± 1.71	Sterilized water
74.1 ^b^	19.5 ± 1.91	0 ^b^	0	*Rosa multiflora* Thunb. *Rosa multiflora* Thunb. Ex
72.2 ^b^	19 ± 2.16	0 ^b^	0	*Brassica oleracea* L. Ex
70.3 ^b^	18.5 ± 1.29	0 ^b^	0	*Lactuca serriola* L. Ex
66.5 ^b^	17.5 ± 1.73	0 ^b^	0	*Salvia rosmarinus* Spenn. Ex
56.05 ^c^	14.75 ± 2.75	0 ^b^	0	*Syzygium aromaticum* Ex
56.05 ^c^	14.75 ± 2.5	0 ^b^	0	*Eucalyptus globulus* Labill. Ex
21.34 ^d^	5.5 ± 1.29	0 ^b^	0	*Ocimum basilicum* L. Eo
19.78 ^d^	5.1 ± 1.91	0 ^b^	0	*Pimpinella anisum* L. Eo
15.52 ^d^	4 ± 0.82	0 ^b^	0	*Mentha× piperita* L.Eo

^1^ in pure concentration, ^2^ in 10^−1^ concentration. Values are mean negative efficacy on seed germination of three replications, followed by different letters in the column, which are significantly different (*p* ≤ 0.01).

**Table 5 plants-11-01155-t005:** Effect of essential oils and extracts on the stem and root growth of the seedling *.

Root Growth (mm)	Stem Growth (mm)	Treatment
60.75 ± 4.11	54.5 ± 3.10	Negative control (Sterilized soil)
43 ± 3.36	42 ± 4.08	Positive control (*Pectobacterium betavasculorum*- infested soil)
57.5 ± 2.98	52 ± 2.94	*Mentha× piperita* L.Eo
58 ± 4.57	50 ± 2.44	*Pimpinella anisum* L. Eo
58.5 ± 2.38	50 ± 2.16	*Ocimum basilicum* L. Eo
58 ± 5.74	48.75 ± 3.09	*Rosa multiflora* Thunb. Ex
55.25 ± 3.59	47 ± 2.70	*Lactuca serriola* L. Ex
51.5 ± 3.41	46.5 ± 4.5	*Brassica oleracea* L. Ex
51.5 ± 3.74	46.25 ± 2.87	*Syzygium aromaticum* Ex
51 ± 3.16	46.25 ± 2.62	*Salvia rosmarinus* Spenn. Ex
51.5 ± 4.2	45.75 ± 3.09	*Eucalyptus globulus* Labill. Ex

* There was no statistically significant difference between these essential oils and extracts to improve the seedling’s root and stem growth, Eo = Essential oil, Ex = Plant extract. Values are the average triplicates.

## Data Availability

All the data is available in the manuscript file.

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
