# Peer review of "The Effect of Some Wild Grown Plant Extracts and Essential Oils on Pectobacterium betavasculorum: The Causative Agent of Bacterial Soft Rot and Vascular Wilt of Sugar Beet"

_plants, 2022, doi:10.3390/plants11091155_

Round 1

Reviewer 1 Report

Reviewer#1

The manuscript is well written and presented excellent information. Authors analyzed the various concentrations and combinations of the different essential oils and plant extracts for antibacterial effect on Pb exposure plants with effects on seed germination and seedling growth. I strongly recommend, the manuscript can be accept after minor changes/corrections mentioned below:

  1. Table 2 author may separate for EO and EX analysed plant samples?

Why did you analysed EO for few plants and Ex for few samples separately.

  1. Discussion should include more (literatures) and needed improvement
  2. Please mention what are the positive control and negative control used in this study (figure caption and table legends too)

Reviewer 1 Report

The manuscript is well written and presented excellent information. Authors analyzed the various concentrations and combinations of the different essential oils and plant extracts for antibacterial effect on Pb exposure plants with effects on seed germination and seedling growth. I strongly recommend, the manuscript can be accept after minor changes/corrections mentioned below:

Reviewer#1

We would like to thank to Reviewer 1 valuable comments. We did all the changes you mentioned in yellow highlight through the paper.

Q1) Table 2 author may separate for EO and EX analyzed plant samples?

A1) These two tables were separated.

Q2) Why did you analyze EO for few plants and Ex for few samples separately?

A2) We wanted to evaluate the antibacterial effect of Eo and Ex on bacterial growth and finally we could determine the best effective essential oils and plant extracts on Pectobacterium betavasculorum.  

Q3) Discussion should include more (literatures) and needed improvement.

A3) It was done in yellow highlight color.

Q4) Please mention what are the positive control and negative control used in this study (figure caption and table legends too).

A4) It was done in yellow highlight color.

Reviewer 2 Report

The manuscript redacted by Rastgou and co-workers is a good example of huge and time-consuming work presented in a light and easy to understand way. However, poorly argued in regard to greener/sustainable approaches...

There are a lot of missing italic forms for bacteria and plant names, "in vitro"... Please take another look at these details.

L70: add refs, please

Do the authors quantify the extraction yields? If you did, please add.

Table 1. It would be better a list of the used plants and plant parts (column 1) with a simple mark to signal if it was addressed hydrodistillation (column 2), soxhlet (column 3) or both. It will be easier to quickly identify which results can be compared.

L82: use numbers to express the amount of material (as previously in the same section), please.

L86: please avoid personal statements as "we". please rephrase this sentence

L99: "NA" please, use acronyms just before explaining them

Section 2.1: please subdivide hydrodistillation and soxhlet procedures with a simple paragraph. it will be easier to understand that they are two different methodologies of extraction

Section 2.3: "nutrient glucose agar", which one? please add composition and suppliers

L118: please correct the concentration properly

L152: probably it is about table 2, am I right?

Table 2: what is the meaning of a diameter of 0mm at level "c" of significant difference?

Table 3: "Negative efficacy", can you better explain what you want to report with this column, please?

Table 4: no statistical analysis is mentioned. If no differences were found, please add this info as a table footnote.

Do the authors have the chance of analysing the obtained extracts by GC?

Do the authors have a clue of which are the main molecules with the ability to achieve these results?

Do the authors use any positive control such as pure terpenes as standard? Figure 1 mentioned a "positive control", however, I'm not finding descriptions in section 2.

Section 4: well done but, please, add a conclusion section as well.

Reviewer 2 Report

The manuscript redacted by Rastgou and co-workers is a good example of huge and time-consuming work presented in a light and easy to understand way. However, poorly argued in regard to greener/sustainable approaches...

We would like to thank to Reviewer 2 for valuable comments. We did all the changes you mentioned in light blue highlight through the paper.

Q1) There are a lot of missing italic forms for bacteria and plant names, "in vitro"... Please take another look at these details.

A1) We spotted all of them and changed them in light blue highlight color.

Q2) L70: add refs, please.

A2) Reference No.23 was added.

Q3) Do the authors quantify the extraction yields? If you did, please add.

A3) No we didn’t.

Q4) Table 1. It would be better a list of the used plants and plant parts (column 1) with a simple mark to signal if it was addressed hydrodistillation (column 2), soxhlet (column 3) or both. It will be easier to quickly identify which results can be compared.

A4) All of the essential oils were extracted by hydro-distillation and all of the plant extracts were extracted by soxhlet. as mentioned in the text.

Q5) L82: use numbers to express the amount of material (as previously in the same section), please.

A5) It was done.

Q6) L86: please avoid personal statements as "we". please rephrase this sentence.

A6) The sentence was rephrased.

Q7) L99: "NA" please, use acronyms just before explaining them.

A7) It was replaced by the full name; nutrient agar.

Q8) Section 2.1: please subdivide hydrodistillation and soxhlet procedures with a simple paragraph. it will be easier to understand that they are two different methodologies of extraction.

A8) These two methods were separated in this section.

Q9) Section 2.3: "nutrient glucose agar", which one? please add composition and suppliers.

A9) Nutrient agar with composition and supplier was added.

Q10) L118: please correct the concentration properly.

A10) It was done.

Q11) L152: probably it is about table 2, am I right?

A11) Yes. The table one separated into two tables and so the Table 2 replaced by Table 3.

Q12) Table 2: what is the meaning of a diameter of 0mm at level "c" of significant difference?

A12) In pure concentration of essential oils and extracts, no seed germinated.

 Q13) Table 3: "Negative efficacy", can you better explain what you want to report with this column, please?

 A13) The explanation was added to the text. In preliminary experiments, it was indicated that several used essential oils and extracts caused phytotoxicity and inhibited seeds germination. For removing these negative effects as well as statistical analysis with low errors, the negative efficacy of the essential oils and extracts were measured.

Q14) Table 4: no statistical analysis is mentioned. If no differences were found, please add this info as a table footnote.

A14) It was added as a footnote to the table.

Q15) Do the authors have the chance of analyzing the obtained extracts by GC?

A15) Unfortunately, we didn’t have enough grant to do this part, but we plan to do it in the future.

Q16) Do the authors have a clue of which are the main molecules with the ability to achieve these results?

A16) We didn’t use GC to determine the composition of these essential oils and extracts, but as it mentioned in the paper, some of these essential oils and extracts such as Tymus vulgaris has stronger antimicrobial properties as its oil poses a higher phenolic material, including carvacrol, eugenol, and thymol.

Q17) Do the authors use any positive control such as pure terpenes as standard? Figure 1 mentioned a "positive control", however, I'm not finding descriptions in section 2.

A17) Unfortunately, we didn’t have pure terpenes as standard. Glass containers filled by sterilized soil and sterilized seeds used as negative control and glass containers filled by Pectobacterium betavasculorum-infested soil and sterilized seeds used as a positive control.

Q18) Section 4: well done but, please, add a conclusion section as well.

A18) Conclusion was added to this section

Reviewer 3 Report

Dear Editor,

 the work entitled "The effect of some wild grown plant extracts on Pectobacterium betavasculorum: the causative agent of bacterial soft rot and vascular wilt of sugar beet" has the purpose of evaluating the antimicrobial activity of 27 essential oils against Pectobacterium betavasculorum because until now there are no data of the antimicrobial activity of these 27 oils against this fungus. It is for this reason that even if the authors have not yet carried out a quali-quantitative analysis of the composition of the oils under study, the work can be considered for publication. This work can be considered as a preliminary work. However for the publication is necessary a little revision.

Abstract

-The name of all plants must be followed by the author's name

page 1, line 29: Pimpinella anisum must be written in Italics

Materials and Methods

page 3, line 97-105: explain well the method of standardization of the inoculum. Insert one or more references

Reviewer 3 Report

The work entitled "The effect of some wild grown plant extracts on Pectobacterium betavasculorum: the causative agent of bacterial soft rot and vascular wilt of sugar beet" has the purpose of evaluating the antimicrobial activity of 27 essential oils against Pectobacterium betavasculorum because until now there are no data of the antimicrobial activity of these 27 oils against this fungus. It is for this reason that even if the authors have not yet carried out a quali-quantitative analysis of the composition of the oils under study, the work can be considered for publication. This work can be considered as a preliminary work. However for the publication is necessary a little revision.

Reviewer#3

We would like to thank to Reviewer 3 valuable comments. We did all the changes you mentioned in green highlight through the paper.

Q1) Abstract: The name of all plants must be followed by the author's name

A1) It was done.

Q2) page 1, line 29: Pimpinella anisum must be written in Italics

A2) It was done.

Q3) Materials and Methods

A3) It was not clear what is the suggestion of reviewer, so we can not response.

Q4) page 3, line 97-105: explain well the method of standardization of the inoculum. Insert one or more references

A4) The sentences and references were added.

Q5) L70: add refs, please

A5) It was done.

Round 2

Reviewer 2 Report

The authors greatly revised the manuscript improving its quality. Therefore, in my opinion, it can be published in the present form.

Author Response

The authors are thankful to the reviewer for such an excellent review of the paper
